# Avoiding false discoveries in single-cell RNA-seq by revisiting the first Alzheimer's disease dataset

**Alan E Murphy[1,2]\*, Nurun Fancy[1,2], Nathan Skene[1,2]\***

[1]UK Dementia Research Institute at Imperial College London, London, United Kingdom; [2]Department of Brain Sciences, Imperial College London, London, United Kingdom

**Abstract** Mathys et al. conducted the first single-nucleus RNA-seq (snRNA-seq) study of Alzheimer's disease (AD) (Mathys et al., 2019). With bulk RNA-seq, changes in gene expression across cell types can be lost, potentially masking the differentially expressed genes (DEGs) across different cell types. Through the use of single-cell techniques, the authors benefitted from increased resolution with the potential to uncover cell type-specific DEGs in AD for the first time. However, there were limitations in both their data processing and quality control and their differential expression analysis. Here, we correct these issues and use best-practice approaches to snRNA-seq differential expression, resulting in 549 times fewer DEGs at a false discovery rate of 0.05. Thus, this study highlights the impact of quality control and differential analysis methods on the discovery of disease-associated genes and aims to refocus the AD research field away from spuriously identified genes.

**\*For correspondence:**
alanmurph94@hotmail.com
(AEM);
n.skene@imperial.ac.uk (NS)

**Competing interest:** The authors declare that no competing interests exist.

## eLife assessment

This paper reports a **useful** finding on the impact of choices of quality control and differential analysis methods on the discovery of disease-associated gene expression signatures. The study provides a **solid** comparison of the data process by re-analysis of a large-scale snRNA-seq dataset for Alzheimer's disease. This paper would be of interest to the community as to rigorous analyses for large-scale single-cell datasets.

## Introduction

*Mathys et al., 2019* undertook the first single-nucleus RNA-seq (snRNA-seq) study of Alzheimer's disease (AD). The authors profiled the transcriptomes of approximately 80,000 cells from the prefrontal cortex, collected from 48 individuals – 24 of which presented with varying degrees of AD pathology. (*Mathys et al., 2019*) data processing and quality control (QC) strategy for their snRNA-seq data was state of the art at this time. Furthermore, the authors took extra measures in an attempt to ensure the reliability of their results. Here, we reanalyse this data as not a criticism of the study, but as an endeavour to raise awareness and provide recommendations for rigorous analysis of single-cell and single-nucleus RNA-seq data (sc/snRNA-seq) for future studies. Most importantly, we aim to ensure that the AD research field does not focus on spuriously identified genes.

## Results and discussion

Our questions of *Mathys et al., 2019* focus around their data processing and their differential expression (DE) analysis (*Figure 1*). Firstly, in relation to their processing approach, the authors discussed

**Figure 1.** Pseudobulk differential expression results in far less dubious disease-related genes. (**a, b**) The log$_2$ fold change and -log$_{10}$ false discovery rate (FDR) of the differentially expressed genes (DEGs) from the authors' original work (Mathys et al.) and our reanalysis (Our analysis). In (**b**), we have marked an FDR of $5 \times 10^{-7}$, dashed grey line, to highlight how small the p-values from Mathys et al.'s analysis are. For (**a, b**), n is based on the number of DEGs: 26 for our analysis and 23,923 for Mathys et al. (**c–g**) show the Pearson correlation between the cell counts after quality control (QC) and the number of DEGs identified - n is the 6 cell types tested. For (**f, g**) analysis, the samples have been randomly mixed between case and control patients - n = 100 random permutations. The different cell types are astrocytes (Astro), excitatory neurons (Exc), inhibitory neurons (Inh), microglia (Micro), oligodendrocytes (Oligo), and oligodendrocyte precursor cells (OPC).

the high percentages of mitochondrial reads and low number of reads per cell present in their data. This is indicative of low cell quality (*Ilicic et al., 2016*); however, we believe the authors' QC approach may not capture all of these low-quality cells. Moreover, the authors did not integrate the cells from different individuals to account for batch effects. As the field has matured since the authors' work was published, dataset integration has become a common step in sc-RNA-seq protocols and is recommended by some to remove confounding sources of variation (*Heumos et al., 2023*; *Amezquita et al., 2020*; *Tran et al., 2020*). To gain advantage of these recent approaches, we used scFlow (*Khozoie et al., 2021*) to reprocess the authors' data. This pipeline included the removal of empty droplets, nuclei with low read counts and doublets, followed by embedding and integration of cells from separate samples and cell typing. scFlow combines best-practice approaches for processing sc/snRNA-seq datasets; see 'Materials and methods' for a detailed explanation of these steps. Reprocessing resulted in 50,831 cells passing QC, approximately 20,000 less than the authors' postprocessing set with differing cell-type proportions (*Figures 2 and 3*).

With regards to data quality, it is worth noting that over 99% of nuclei had less than 200 genes expressed (*Table 1*). While this QC step was not unique to our reprocessing, the authors made the same exclusion in their analysis (*Mathys et al., 2019*), it highlights the relatively low quality of the data which may be attributable to the early stage of snRNA-seq technology of the time. For example, Brase et al.'s recent study of snRNA-seq of autosomal-dominant AD (*Brase et al., 2023*) used a more stringent cut-off for the minimum number of expressed genes and still kept 27% (122 times more) of the assayed cells after all QC steps. Moreover, the authors discussed the high percentages of mitochondrial reads in their data. The differences in approaches to filtering

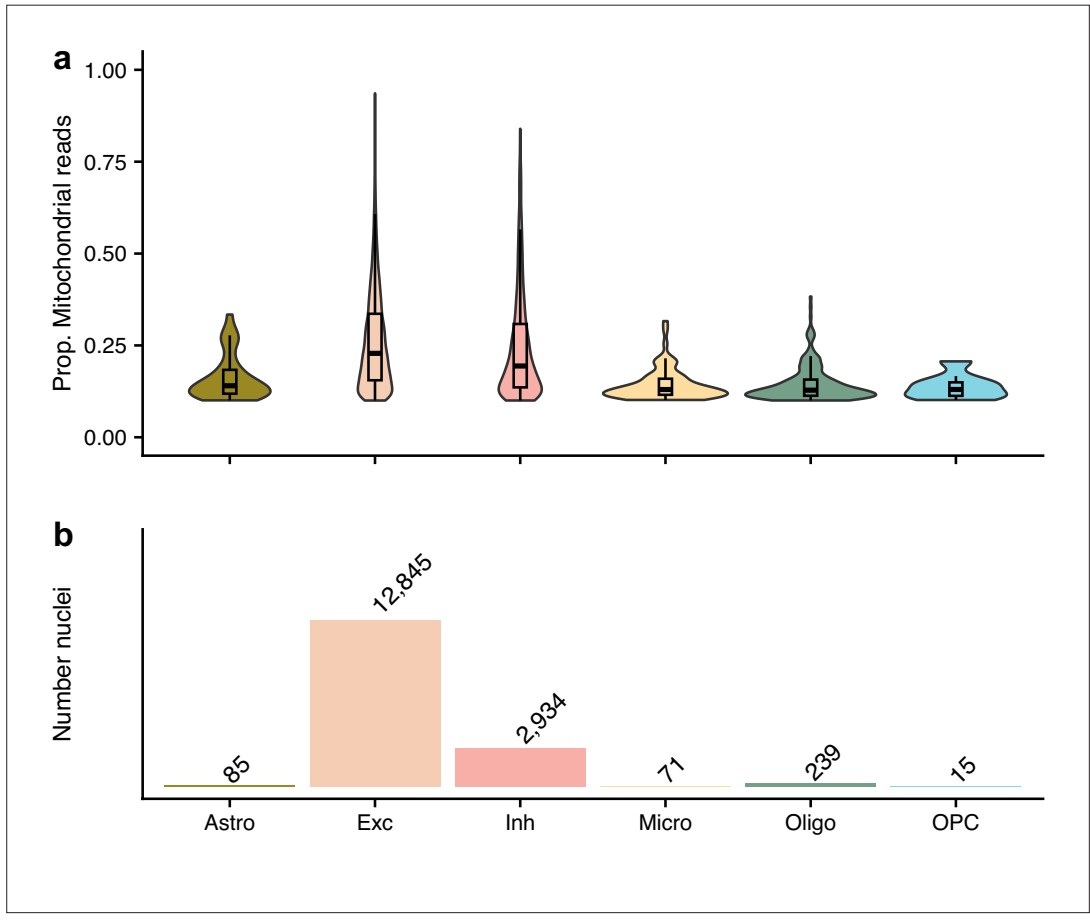

**Figure 2.** The nuclei that were removed from our quality control approach as their proportion of mitochondrial reads were ≥10%, but kept in the authors'. (**a**) shows the proportion of mitochondrial reads across the different cell types. (**b**) gives the number of removed nuclei which were kept by the authors. The different cell types are astrocytes (Ast), excitatory neurons (Ex), inhibitory neurons (In), microglia (Mic), oligodendrocytes (Oli), and oligodendrocyte precursor cells (Opc).

**Table 1.** Overview of the aggregated number of cells across samples removed at each step of the quality control (QC) as part of scFlow.

Note that cells can fail QC for more than one check, so only the total failed and total passed rows will sum to 100%.

| QC steps | Total cells | Percentage |
|---|---|---|
| Pre-QC | 35,389,440 | |
| Total failed | 35,337,874 | 99.85 |
| Minimum library size (n < 200) | 35,307,281 | 99.77 |
| Maximum library size | 4742 | 0.01 |
| Minimum expressed genes (n < 200) | 35,312,434 | 99.78 |
| Maximum library size/expressed genes (MAD> 4) | 2149 | 0.01 |
| Proportion of mitochondrial genes (≥ 0.1) | 1,097,738 | 3.10 |
| Multiplets (pK = 0.0054) | 581 | 0.00 |
| Total passed | 51,566 | 0.15 |

MAD, median absolute deviation.

based on the proportion mitochondrial reads accounts for the notable discrepancy in the number of nuclei after QC between our approach and the authors'. Our approach used a 10% cut-off for the proportion of mitochondrial reads in a nuclei, as set out in Amezquita et al.'s best-practice guidelines (*Amezquita et al., 2020*), which is less stringent than Seurat's guidelines (5%) (*Hao et al., 2021*) or that from *Heumos et al., 2023* (8% from a median absolute deviations [MAD]-based cut-off selection). Conversely, the authors filtered out high mitochondrial read nuclei based on clusters from their t-SNE projection of the data (*Mathys et al., 2019*). Even at our lenient cut-off, over 16,000 nuclei that were removed in our QC pipeline were kept by the authors' *Figure 2*, explaining the discrepancy in the number of nuclei after QC. Based on *Figure 2*, it is clear that the authors'

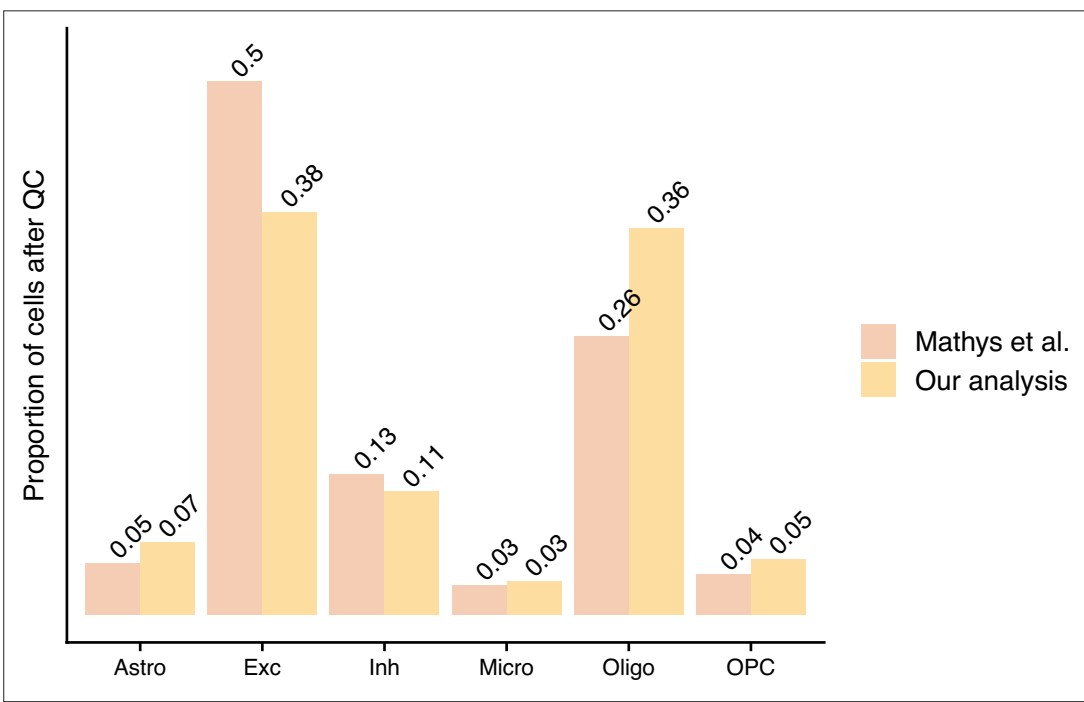

**Figure 3.** The proportion of cells left after quality control (QC) from the authors' processing approach (Mathys et al.) and our standardised pipeline approach – scFlow (Our analysis).

approach was ineffective at removing nuclei with high proportions of mitochondrial reads which is indicative of cell death (*Heumos et al., 2023*; *Ilicic et al., 2016*) – both excitatory and inhibitory nuclei with higher than 75% reads from the mitochondria were kept in the final processed dataset by the authors. We have made the data from our alternative processing approach publicly available (through Synapse: https://doi.org/10.7303/syn51758062.1) so that researchers can utilise this resource free of low-quality nuclei.

Our second question of *Mathys et al., 2019* is their DE approach. The authors conducted a DE analysis between the controls and the patients with AD pathology, concentrating on six neuronal and glial cell types; excitatory neurons, inhibitory neurons, astrocytes, microglia, oligodendrocytes, and oligodendrocyte precursor cells, derived from the Allen Brain Atlas (*Tasic et al., 2018*). They performed downstream analysis on their identified differentially expressed genes (DEGs) and investigated some of the most compelling genes in more detail. Therefore, all findings put forward by their paper were based upon the validity of their DE approach. However, for this approach, the authors conducted a two-part, cell- and patient-level analysis. The cell-level analysis took each cell as an independent replicate, and the results of which were compared for consistency in directionality and rank of their DEGs against their patient-level analysis, a Poisson mixed model. The authors identified 1031 DEGs using this combinatorial approach – DEGs requiring a false discovery rate (FDR) < 0.01 in the cell-level and an FDR < 0.05 in the patient-level analysis. It is important to note that this cell-level DE approach, also known as pseudoreplication, overestimates the confidence in DEGs due to the statistical dependence between cells from the same patient not being considered (*Murphy and Skene, 2022*; *Squair et al., 2021*; *Zimmerman et al., 2021*; *Lazic, 2010*). When we inspect all DEGs identified at an FDR of 0.05 from the authors' cell-level analysis, this number increases to 14,274. Pseudobulk DE analysis has recently been proven to give optimal performance compared to both mixed models and pseudoreplication approaches (*Murphy and Skene, 2022*; *Squair et al., 2021*; *Crowell et al., 2020*; *Soneson and Robinson, 2018*). It aggregates counts to individuals, thus accounting for the dependence between an individual's cells.

Here, to compare the effect of the different DE approaches in isolation, we apply a pseudobulk DE approach (*Chen et al., 2016*) to the authors' original processed data. We found 26 unique DEGs when considering the six cell types used by the authors (*Table 2*). This was 549 times fewer DEGs than that reported originally at an FDR of 0.05. When we compare these DEGs, we can see that the absolute $\log_2$ fold change (LFC) of our DEGs is 15 times larger than the authors'; median LFC of 2.34 and 0.16, despite the authors' DEGs having an FDR score 8000 times smaller; median FDR of $2.89 \times 10^{-7}$ and 0.002 (*Figure 1a and b*). Although we examined a high correlation in the genes' fold change values across our pseudobulk analysis and the authors' pseudoreplication analysis (Pearson R of 0.87 for an adjusted p-value of 0.05, *Table 3*), the p-values and resulting DEGs vary considerably. The correspondence in fold change values is expected given the approaches are applied to the same dataset, whereas the probabilities, which pertain to the likelihood that a gene's expressional changes is related to the case/control differences in AD, importantly do not align. We can show that this stark contrast is just an artefact of the authors taking cells as independent replicates and thus artificially inflating confidence by considering the Pearson correlation between the number of DEGs found and the cell counts (*Figure 1c–e*). There is a near perfect, positive correlation between DEG and cell counts for the authors' pseudoreplication analysis (*Figure 1c*) and for the 1031 genes from the authors' combinatorial approach (*Figure 1d*) which is not present in our pseudobulk reanalysis (*Figure 1e*).

A further point which questions the authors' DE approach is that they identified the vast majority of DEGs in the more abundant, neuronal cell types (*Mathys et al., 2019*). However, an increase in the number of cells is not the same as an increase in sample size since these cells are not independent from one another – they come from the same sample. Therefore, an increase in the number of cells should not necessarily result in an increase in the number of DEGs, whereas an increase in the number of samples would. This point is the major issue with pseudoreplication approaches which overestimate confidence when performing DE due to the statistical dependence between cells from the same patient not being considered (*Squair et al., 2021*; *Lazic, 2010*). In our opinion, it makes more sense to identify the majority of large effect size DEGs in microglia which recent work has established is the primary cell type by which the genetic risk for AD acts (*Skene and Grant, 2016*; *McQuade and Blurton-Jones, 2019*). This is what we found with our pseudobulk DE approach – 96% of all DEGs were in microglia (*Table 2*), whereas only 3% of the authors' DEGs were in microglia.

**Table 2.** The differentially expressed genes from our reanalysis using the same processed data the authors used and pseudobulk differential expression approach.

| Cell | logFC | logCPM | LR | p-Value | adj_pval | HGNC |
|------|-------|--------|-----|---------|----------|------|
| Mic | 2.70178913 | 6.99794619 | 26.1418415 | 3.17E-07 | 0.00061349 | ACRBP |
| Mic | 1.48930071 | 8.06240877 | 28.6361217 | 8.73E-08 | 0.00019303 | APOC1 |
| Mic | 1.09327669 | 8.64199769 | 21.5323014 | 3.48E-06 | 0.00336416 | CD81 |
| Mic | −1.4157681 | 7.93884875 | 23.9955467 | 9.66E-07 | 0.00135806 | CD83 |
| Mic | 3.3782727 | 6.86183548 | 32.0804401 | 1.48E-08 | 4.58E-05 | CLEC1B |
| Mic | 2.84072452 | 6.74370542 | 21.7745509 | 3.07E-06 | 0.00316269 | EGF |
| Mic | 2.55769658 | 6.78345087 | 18.0468872 | 2.16E-05 | 0.01699007 | ELOVL7 |
| Mic | −1.2056098 | 8.33197499 | 22.6644045 | 1.93E-06 | 0.00229576 | IFI44L |
| Mic | −1.6616069 | 7.15366639 | 16.4801274 | 4.92E-05 | 0.03306938 | IFI6 |
| Mic | −1.9809425 | 7.00396289 | 17.9180823 | 2.31E-05 | 0.01699007 | IFIT3 |
| Mic | 2.76502672 | 6.72978805 | 20.6543637 | 5.50E-06 | 0.00472825 | ITGA2B |
| Mic | 1.90963403 | 7.01552233 | 16.3200189 | 5.35E-05 | 0.03448474 | MAP1A |
| Mic | −1.8194508 | 8.26208887 | 45.2221008 | 1.76E-11 | 1.36E-07 | NAMPT |
| Mic | 2.0945044 | 7.11048456 | 20.8068524 | 5.08E-06 | 0.00462318 | NEXN |
| Mic | −2.3789762 | 6.93896985 | 22.3912441 | 2.22E-06 | 0.00245752 | NR4A2 |
| Mic | −2.8553462 | 6.73713862 | 22.8029868 | 1.79E-06 | 0.00229576 | NR4A3 |
| Mic | 3.32873829 | 6.84942721 | 30.955327 | 2.64E-08 | 6.81E-05 | PF4 |
| Mic | 3.4213986 | 6.87326383 | 33.2621657 | 8.05E-09 | 3.11E-05 | PKHD1L1 |
| Mic | 3.64525677 | 6.93422174 | 38.661272 | 5.04E-10 | 2.60E-06 | PPBP |
| Mic | 2.30482679 | 8.10570443 | 60.7932697 | 6.34E-15 | 9.81E-11 | PTPRG |
| Mic | −1.0382468 | 8.11450266 | 15.5968273 | 7.84E-05 | 0.04850839 | RORA |
| Mic | 2.54636649 | 6.69202981 | 17.2532606 | 3.27E-05 | 0.02300507 | SDPR |
| Mic | −0.9629617 | 8.8434334 | 17.9319131 | 2.29E-05 | 0.01699007 | SYTL3 |
| Mic | −1.4215374 | 7.99629806 | 25.4736272 | 4.48E-07 | 0.00077092 | TMEM2 |
| Mic | 2.98901596 | 6.77276641 | 24.2100819 | 8.64E-07 | 0.00133637 | TUBB1 |
| Opc | −2.8274718 | 5.03371292 | 22.1334581 | 2.54E-06 | 0.04176231 | EGR1 |

CPM - Counts per Million, LR - fold change ratio, HGNC - HUGO Gene Nomenclature Committee.

Although it has been proven that pseudoreplication approaches result in false positives by artificially inflating the confidence from non-independent samples, we wanted to investigate the effect of the approach on the authors' dataset. We ran the same cell-level analysis approach – a Wilcoxon rank-sum test and FDR multiple-testing correction – 100 times whilst randomly permuting the patient identifiers (*Figure 1f*). We would expect to find minimal DEGs with this approach given the random mixing of case and control patients. However, this pseudoreplication approach consistently found high numbers of DEGs, and we observe the same correlation between the number of cells and the number of DEGs as with the authors' results. We did not observe the same pattern when running the same analysis with pseudobulk DE (*Figure 1g*). As a result, we conclude that integrating this pseudoreplication approach with a mixed model like the authors proposed just artificially inflates the test confidence for a random sample of the genes resulting in more false discoveries in cell types with bigger counts.

Up to this point, to compare the effect of the DE approaches in isolation, we analysed the same processed data from the authors as opposed to our reprocessed data. We also performed pseudobulk

**Table 3.** Pearson correlation between our pseudobulk differential expression analysis and the authors' pseudoreplication analysis on all genes found to be significant at different adjusted p-value cut-offs from the authors' pseudoreplication analysis.

| Pseudoreplication adjusted p-value cut-off | Number of genes compared | Pearson correlation |
|---|---|---|
| 0.01 | 20,152 | 0.8646269 |
| 0.05 | 23,903 | 0.8708275 |
| 0.1 | 26,382 | 0.8721126 |
| 0.25 | 32,117 | 0.8764692 |
| 0.5 | 42,022 | 0.8751554 |
| 1 | 84,467 | 0.826248 |

DE on our reprocessed data and found 16 unique DEGs (*Table 4*). It is worth noting that the fold change correlation between our two DE analyses (reprocessed data vs authors' processed data) on the identified DEGs is only moderate (Pearson R of 0.57) and is lower than that of the correlation between pseudoreplication and pseudobulk on the same dataset (*Table 3*). This highlights the effect that the low quality high mitochondrial read cells have on DE analysis.

In conclusion, the authors' analysis has been highly influential in the field with numerous studies undertaken based on their results, something we show has uncertain foundations. However, we would like to highlight that the use of pseudoreplication in neuroscience research is not isolated to the authors' work; others have used this approach (*Fernandes et al., 2020*; *Lui et al., 2021*; *Wakhloo et al., 2020*), and their results should be similarly scrutinised. Here, we provide our processed count matrix with metadata and also the DEGs identified using an independently validated, DE approach so that other researchers can use this rich dataset free from spurious nuclei or DEGs. While the number of DEGs found here is significantly lower, much greater confidence can be had that these are AD-relevant genes. The low number of DEGs found may also cause concern given the sample size and cost of collection and sequencing of such datasets. However, the increasing number of snRNA-seq studies being conducted for AD creates the opportunity to conduct differential meta-analyses to increase power. Further work is required in the field to develop methods to conduct such analysis, integrating studies and accounting for their

**Table 4.** The differentially expressed genes from our reanalysis using the reprocessed data and pseudobulk differential expression approach.

| Cell | logFC | logCPM | LR | p-Value | adj_pval | ensembl_id | HGNC |
|---|---|---|---|---|---|---|---|
| OPC | –4.1544663 | 4.92100803 | 21.6911445 | 3.20E-06 | 0.04985906 | ENSG00000166573 | GALR1 |
| Astro | –4.5845276 | 4.7965143 | 22.2367847 | 2.41E-06 | 0.037634 | ENSG00000137959 | IFI44L |
| Micro | –3.7616619 | 7.32875316 | 26.8149688 | 2.24E-07 | 0.00077905 | ENSG00000077238 | IL4R |
| Micro | –2.0681446 | 7.88736441 | 17.5929095 | 2.74E-05 | 0.0346187 | ENSG00000105835 | NAMPT |
| Micro | –1.6757556 | 7.58472506 | 19.1736829 | 1.19E-05 | 0.02076348 | ENSG00000118257 | NRP2 |
| Micro | –3.1556403 | 6.85232653 | 19.2064627 | 1.17E-05 | 0.02076348 | ENSG00000135363 | LMO2 |
| Micro | –3.4339265 | 6.9290472 | 19.5975589 | 9.56E-06 | 0.02076348 | ENSG00000138135 | CH25H |
| Micro | –2.8183109 | 6.77500676 | 16.907959 | 3.92E-05 | 0.04550806 | ENSG00000142408 | CACNG8 |
| Micro | 2.90076647 | 8.34560617 | 45.5144266 | 1.52E-11 | 2.11E-07 | ENSG00000144724 | PTPRG |
| Micro | 3.25867589 | 6.91671013 | 16.5519147 | 4.73E-05 | 0.0490155 | ENSG00000163106 | HPGDS |
| Micro | –2.0290905 | 7.12321166 | 16.4746746 | 4.93E-05 | 0.0490155 | ENSG00000171612 | SLC25A33 |
| Micro | –3.4657301 | 6.93307221 | 19.7883301 | 8.65E-06 | 0.02076348 | ENSG00000172243 | CLEC7A |
| Micro | –4.172807 | 7.16813583 | 34.3515807 | 4.60E-09 | 3.20E-05 | ENSG00000174600 | CMKLR1 |
| Micro | –3.1984588 | 6.87310555 | 18.5335889 | 1.67E-05 | 0.0232342 | ENSG00000227531 | RP11-202G18.1 |
| Micro | 3.40562887 | 6.9381703 | 18.5526502 | 1.65E-05 | 0.0232342 | ENSG00000228058 | RP11-552D4.1 |
| Micro | 4.46073301 | 7.66559163 | 29.7716679 | 4.86E-08 | 0.00022549 | ENSG00000253496 | RP11-13N12.1 |

heterogeneity, similar to that which has been done for bulk RNA-seq (*Rau et al., 2014*). Some such approaches have already been made in COVID-19 research which could be leveraged for neurode-generative disease (*Garg et al., 2021*).

# Materials and methods
## Processing of sc/snRNA-seq dataset
The data reprocessing was conducted with scFlow (*Khozoie et al., 2021*), the steps of which are discussed in the following two sections.

### Quality control of snRNAseq data
The raw snRNA-seq data (10.7303/syn18485175) and the ROSMAP metadata (10.7303/syn3157322) were downloaded from https://www.synapse.org/ upon acquiring appropriate approval. Downstream primary analyses of gene–cell matrices were performed using our scFlow pipeline (*Khozoie et al., 2021*). To determine ambient RNA profile and distinguish true nuclei from empty droplets, empty-Drops was used with a lower parameter of <100 counts, an alpha cut-off of ≤0.001, and with 10,000 Monte Carlo iterations (*Lun et al., 2019*). This approach has been recommended as best practice in the literature (*Amezquita et al., 2020*). Nuclei were then filtered for ≥200 total counts and ≥200 total expressed genes, which was defined as a minimum of 2 counts in at least three cells. We excluded any nuclei with total counts or total expressed genes with more than 4 MAD defined by an adaptive thresholding method. Nuclei were excluded if the proportion of counts mapping to mitochondrial genes was more than 10%, as set out in best-practice guidelines (*Amezquita et al., 2020*). Doublets were identified using the DoubletFinder algorithm, with a doublets-per-thousand-cells increment of eight cells (recommended by 10X Genomics), and a pK value of 0.005 (*McGinnis et al., 2019*). DoubletFinder was shown to be the best overall performing method in a recent benchmark (*Xi and Li, 2021*). The aggregated number of cells and proportions dropped at each step is given in *Table 1* while a comparison of the proportion of cells in each cell type after reprocessing compared to the authors' processed data is given in *Figure 2*. All files from the scFlow run, including QC statistics, are available in the GitHub repository in the scFlow_files folder (copy archived at *Murphy, 2023*). This includes sample-level genes and cells' QC numbers.

### Integration and clustering
The linked inference of genomic experimental relationships (LIGER) package was used to calculate integrative factors across samples (*Welch et al., 2019*). LIGER was recently found to be one of the top performing methods for batch-effect correction (*Tran et al., 2020*). LIGER parameters used included k: 30; lambda: 5.0; thresh: 0.0001; max_iters: 100; knn_k: 20; min_cells: 2; quantiles: 50; nstart: 10; resolution: 1; num_genes: 3000; and centre: false. Two-dimensional embeddings of the LIGER integrated factors were calculated using the Uniform Manifold Approximation and Projection (UMAP) algorithm with the following parameters: pca_dims: 50; n_neighbours: 35; init: spectral; metric: euclidean; n_epochs: 200; learning_rate: 1; min_dist: 0.4; spread: 0.85; set_op_mix_ratio: 1; local connectivity: 1; repulsion_strength: 1; negative_sample_rate: 5; and fast_sgd: false (*McInnes et al., 2020*). The Leiden community detection algorithm was used to detect clusters of cells from the 2D UMAP (LIGER) embeddings; a resolution parameter of 0.001 and a k value of 50 were used (*Traag et al., 2019*). This approach has been noted as best practice by a recent review (*Heumos et al., 2023*). Automated cell typing of the detected clusters was performed as previously described using the Expression Weighted Celltype Enrichment algorithm in scFlow against a previously generated cell-type data reference from the Allen Human Brain Atlas (*Hodge et al., 2019*; *Skene and Grant, 2016*). The top five marker genes for each automatically annotated cell type were determined using Monocle 3 and validated against canonical cell-type markers (*Trapnell et al., 2014*).

## DE analysis
All DE analyses were conducted using pseudobulk DE approach with sum aggregation and edgeR LRT (*Chen et al., 2016*). Pseudobulk aggregates nuclei within a biological replicate (an individual) for each cell type, reducing the dropout issue in single-cell data and avoiding the false inflation of confidence from non-independent samples of pseudoreplication approaches (*Squair et al., 2021*; *Murphy and*

*Skene, 2022*). The DE analysis pipeline is available at GitHub repository (copy archived at *Murphy, 2023*). This is a general use pipeline which can be run for any single-nucleus or single-cell transcriptomic dataset. Note that we report DEGs across AD and controls using the same processed data the authors used (*Table 2*) and using our reprocessed data (*Table 4*).

### Code availability

The DE analysis pipeline is available at GitHub repository (copy archived at *Murphy, 2023*). This is a general use pipeline which can be run for any single-nucleus or single-cell transcriptomic dataset. The config file containing all the parameters used and QC overview file for the scFlow run is also available in this repository.

## Acknowledgements

This work was supported by a UKDRI Future Leaders Fellowship (grant number MR/T04327X/1) and the UK Dementia Research Institute, which receives its funding from UK DRI Ltd, funded by the UK Medical Research Council, Alzheimer's Society and Alzheimer's Research UK. The results published here are in whole or in part based on data obtained from the AD Knowledge Portal (https://adknowledgeportal.org). The data available in the AD Knowledge Portal would not be possible without the participation of research volunteers and the contribution of data by collaborating researchers. Study data were provided by the Rush Alzheimer's Disease Center, Rush University Medical Center, Chicago. Data collection was supported through funding by NIA grants P30AG10161 (ROS), R01AG15819 (ROSMAP; genomics and RNAseq), R01AG17917 (MAP), R01AG30146, R01AG36836 (RNA-seq), U01AG32984 (genomic and whole-exome sequencing), U01AG46152, U01AG61356 (ROSMAP AMP-AD, targeted proteomics), U01AG46161 (TMT proteomics), U01AG61356 (whole genome sequencing, targeted proteomics, ROSMAP AMP-AD), the Illinois Department of Public Health (ROSMAP), and the Translational Genomics Research Institute (genomic). Additional phenotypic data can be requested at https://www.radc.rush.edu/.

## Additional information

### Funding

| Funder | Grant reference number | Author |
|---|---|---|
| UK Research and Innovation | Future Leaders Fellowship (MR/T04327X/1) | Nathan Skene |
| UK Dementia Research Institute | | Nathan Skene |

The funders had no role in study design, data collection and interpretation, or the decision to submit the work for publication.

### Author contributions

Alan E Murphy, Conceptualization, Resources, Data curation, Software, Formal analysis, Validation, Investigation, Visualization, Methodology, Writing – original draft; Nurun Fancy, Data curation, Software, Formal analysis, Validation; Nathan Skene, Conceptualization, Supervision, Methodology, Project administration, Writing - review and editing

### Author ORCIDs

Alan E Murphy  https://orcid.org/0000-0002-2487-8753
Nurun Fancy  http://orcid.org/0000-0002-6481-6266

Joint Public Review: https://doi.org/10.7554/eLife.90214.3.sa1
Author Response https://doi.org/10.7554/eLife.90214.3.sa2

## Additional files

### Supplementary files
• MDAR checklist

### Data availability
The differentially expressed genes and processed count matrix from the original study are available with their manuscript. The count matrix and metadata from our reprocessing approach are available via the AD Knowledge Portal (https://adknowledgeportal.org). The AD Knowledge Portal is a platform for accessing data, analyses, and tools generated by the Accelerating Medicines Partnership (AMP-AD) Target Discovery Program and other National Institute on Aging (NIA)-supported programs to enable open-science practices and accelerate translational learning. The data, analyses, and tools are shared early in the research cycle without a publication embargo on secondary use. Data is available for general research use according to the following requirements for data access and data attribution (https://adknowledgeportal.org/DataAccess/Instructions). For access to content described in this article, see https://doi.org/10.7303/syn51758062.1. All other relevant scripts and data for working with this dataset and supporting the key findings of this study are available within the article or from our GitHub repository (copy archived at *Murphy, 2023*).

The following previously published dataset was used:

| Author(s) | Year | Dataset title | Dataset URL | Database and Identifier |
|---|---|---|---|---|
| Mathys H, Davila-Velderrain J, Peng Z, Gao F, Mohammadi S, Young JZ, Menon M, He L, Abdurrob F, Jiang X, Martorell AJ, Ransohoff RM, Hafler BP, Bennett DA, Kellis M, Tsai LH | 2019 | Single-cell transcriptomic analysis of Alzheimer's disease | https://www.synapse.org/#!Synapse:syn18485175 | Synapse, 10.7303/syn18485175 |

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
