## [Editor Report · eLife assessment]

This paper reports a **useful** finding on the impact of choices of quality control and differential analysis methods on the discovery of disease-associated gene expression signatures. The study provides a **solid** comparison of the data process by re-analysis of a large-scale snRNA-seq dataset for Alzheimer's disease. This paper would be of interest to the community as to rigorous analyses for large-scale single-cell datasets.

---

## [Referee Report · Joint Public Review]

Murphy, Fancy and Skene performed a reanalysis of snRNA-seq data from Alzheimer Disease (AD) patients and healthy controls published previously by Mathys et al. (2019), arriving at the conclusion that many of the transcriptional differences described in the original publication were false positives. This was achieved by revising the strategy for both quality control and differential expression analysis. With this re-analysis, the authors aim to raise awareness of the impact of data analysis choices for scRNA-seq data and to caution focus on putatively wrongly identified genes in the AD research community. The revised manuscript has been improved by separating QC and DE analysis, which makes interpretation of both steps more straightforward.

STRENGTHS:

The authors demonstrate that the choice of data analysis strategy can have a vast impact on the results of a study, which in itself may not be obvious to many researchers.

The authors apply a pseudobulk-based differential expression analysis strategy (essentially, adding up counts from all cells per individual and comparing those counts with standard RNA-seq differential expression tests), which is (a) in line with latest community recommendations, (b) different from the "default options" in most popular scRNA-seq analysis suites, and (c) explains the vastly different number of DEGs identified by the authors and the original publication. The recommendation of this approach together with a detailed assessment of the DEGs found by both methodologies could potentially be a useful finding for the research community. Unfortunately, it is currently not sufficiently substantiated.

All code and data used in this study are publicly available to the readers.

WEAKNESSES:

The authors interpret the fact that they found fewer DEGs with their method than the original paper as a good thing by making the assumption that all genes that were not found were false positives. However, they do not prove this, and it is likely that at least some genes were not found due to a lack of statistical power and not because they were actually "incorrect". The original paper also had performed independent validations of some genes that were not found here. I had raised this weakness in my first review, but it was not explicitly addressed and still pertains to the revised manuscript. The authors have added an analysis that shows that "pseudoreplication" is prone to false positive (FP) discoveries for high cell numbers (Fig. 1f), but this does not prove that all of Mathys' DEGs were wrong.

I am concerned that almost all DEGs found by the authors are in the rare cell types, foremost the rare microglia (see Fig. 1e). Indeed, there is a weak negative correlation between cell counts and numbers of DEGs (Fig. 1e), if the correlation analysis is to be believed (see next point). It is unclear to me how many cells the pseudo-bulk counts were based on for these cell types, but it seems that (a) there were few and (b) there were quite few reads per cells. If both are the case, the pseudobulk counts for these cell populations might be rather noisy and the DEG results liable to outliers with extreme fold changes. Supp. Fig. 3b now shows three examples of DEGs, of which one (EGR1) looks like the DE call is indeed largely driven by four outliers, while Supp. Fig 3a shows at least one gene (BEX1) that could be FP of the pseudobulk approach due to insufficient statistical power. The authors go on to cite two papers (one is their own, published in a journal with suspected lack of appropriate quality assurance measures https://predatoryreports.org/the-predatory-journals-1), to support that the finding of DEGs in microglia "makes more sense" (l. 127). In summary, neither the presented examples nor the supporting literature are convincing. Lastly, the authors even show themselves that their approach is liable to FPs if applied with very low cell numbers in the range of those for microglia and OPCs (Fig. 1g).

The correlation analysis between cell counts and number of DEGs found is weak. In all three cases (Fig. 1c, d, e) the correlation is largely driven by a single outlier data point.

The authors claim they improved the quality control of the dataset but offer no objective metric to assess this putative improvement. The authors' QC procedure removes some 20k cells that had not been filtered out by Mathys' et al. As the authors state themselves, this difference is mostly due to the removal of cells with a high mitochondrial read content. Murphy et al use a fixed threshold for the mitochondrial percentage of reads, while the original paper had removed cell clusters with an "abnormally high" mitochondrial read fraction. That also seems reasonable, given that some cells might have a higher mitochondrial read content for reasons other than being "low quality". Simply stating that Mathys' approach was ineffective at removing cells with high mitochondrial read content is a self-fulfilling prophecy given the difference in approach, and itself not proof that the original QC procedure was inferior.

Batch correction: "Dataset integration has become a common step in single-cell RNA-Seq protocols and is recommended to remove confounding sources of variation" (l. 38). While it is true that many authors now choose to perform an integration step as part of their analysis workflow, this is by no means uncontroversial as there is a risk of "over-integration" and loss of true biological differences. I had raised this point previously, but the authors chose not to address it (quoted text and line numbers updated). Given that there is controversy in the literature and "community opinion" on the topic of data integration, this is another example of the authors claiming superiority in analysis without showing proof.

Due to a lack of comparison with other methods and due to the fact that the author's methodology was only applied to a single dataset, the paper presents merely a case study, which could be useful but falls short of providing a general recommendation for a best practice workflow.

APPRAISAL:

The manuscript could help to increase awareness of data analysis choices in the community, but only if the superiority of the methodology was clearly demonstrated. However, the authors only show that there are differences but have no convincing (orthogonal) evidence that their methodology was indeed better. This applies to both QC and DE analysis.

---

## [Author Response]

The following is the authors’ response to the original reviews.

Response to Reviewers

To whom it may concern,Thank you for your constructive feedback on our manuscript. I appreciate the time and effort that you and the reviewers have dedicated to providing your valuable feedback. We are grateful to the reviewers for their insightful comments and suggestions for our paper. I have been able to incorporate changes to reflect the majority of these suggestions provided. I have updated the analysis scripts (at https://github.com/neurogenomics/reanalysis_Mathys_2019) and have listed these changes in blue below:

eLife assessment:This work is useful as it highlights the importance of data analysis strategies in influencing outcomes during differential gene expression testing. While the manuscript has the potential to enhance awareness regarding data analysis choices in the community, its value could be further enhanced by providing a more comprehensive comparison of alternative methods and discussing the potential differences in preprocessing, such as scFLOW. The current analysis, although insightful, appears incomplete in addressing these aspects.

We thank the reviewing editors for this note. We agree that the differences in preprocessing will affect the results and conceal which step in our reanalysis resulted in the discrepancies we noted. To address this, we have split out our reanalysis into two separate parts - In the main body of the text we discuss the differences resulting from just changing the differential expression approach where we use the same processed data as the authors to enable a fair comparison. Secondly, we still provide the reprocessed data and perform differential expression analysis on it and discuss the cause and impact the differences in the processing steps made to the results.

**Reviewer 1:**
I think readers would be interested to learn more about the genes that were found "significant" by the original paper but sorted out by the authors. Did they just fall short of the cutoffs? If so, how many more samples would have been required to ascertain significance? This would yield a recommendation for future studies and an overall more positive/productive spirit to the manuscript. On the other hand, I suspect a fraction of DEGs were false positives due to differences in the proportions of cells from different individuals compared to the original analysis. Which percentage of DEGs does this apply to? Again, this would raise awareness of the issue and support the use of pseudobulk approaches.

To investigate the relationship between the genes and how they differ across our analysis we have added a correlation analysis between our different DE approaches (using the same processed data), see paragraph 5 in the manuscript and supplementary table 3. In short, we find that there is a high correlation in the genes’ fold change values across our pseudobulk analysis and the author’s pseudoreplication analysis on the same dataset (pearson R of 0.87 for an adjusted p-value of 0.05) which is somewhat expected given the DE approaches are applied to the same dataset. However, the p-values, which pertain to the likelihood that a gene’s expressional changes is related to the case/control differences in AD, and resulting DEGs vary considerably due to the artificially inflated confidence of the author’s approach (Fig. 1c-e).Despite there being a correlation between the pseudoreplciation and pseudobulk approaches here, we do not think it makes sense to consider how many more samples would have been required to ascertain significance. The differences in results between the two approaches is not negatable with sample size as many DEGs identified by pseudoreplication will be false positives as highlighted in previous work1,2,3,4.However, perhaps we are misinterpreting the reviewer, who may have meant a power analysis which we have not conducted. Such an undertaking would require analysing a multitude of snRNA-Seq of large sample sizes to garner a confident estimate for power calculations based on pseudobulk approaches. Although we agree with the reviewer that this would be beneficial to the field, we do not believe it is in scope for this work.On the reviewer’s note regarding a fraction of DEGs being false positives due to differences in the proportions of cells from different individuals compared to the original analysis - We have analysed the same processed data the authors used to negate the differences caused by the differing processing steps. We thank the reviewer for this suggestion. We also give more insight into the cause of these differences, namely on filtering our nuclei with large proportions of mitochondrial reads and discuss their effect in paragraph 3 (also see Supplementary Figure 2).

Given there are only a few DEGs, it would be good to show more data about these genes to allow better assessment of the robustness of the results, i.e., boxplots of the pseudobulk counts in the compared groups and perhaps heatmaps of the raw counts prior to aggregation. This could rule out concerns about outliers affecting the results.

In Supplementary Figure 3, we have added boxplots of the sum pseudobulked, trimmed mean of M-values (TMM) normalised counts for three of our identified DEGs (b) and three of the authors’ DEGs which they discuss in their manuscript (a) to show the differences in counts across AD pathology and controls for these genes. We hope this gives some insight into the transcriptional changes highlighted by the differing approaches. In our opinion, there is a clear difference in the transcriptional signal in the genes identified from pseudobulk which is not present for the genes identified from the authors approach.

Overall, I believe the paper would deliver a clearer message by mainlining the QC from the original study and only changing the DE analysis. However, if keeping the part about QC/batch correction:Assess to which degree changes in cell type proportion are indeed due to batch correction (as suggested in the text) and not filtering by looking at the annotated cell types in the original publication and those in your analysis.Also perform the analysis without changing QC and state the # of DEGs in both cases, to at least allow some disentanglement of the effect of different steps of the analysis.Please state the number of cells removed by each QC step in the supplementary note.

We thank the reviewer for this suggestion. We agree with performing the DE analysis on the same processed data as the original authors and have split out our reanalysis into two separate parts, primarily focussing on the discrepancies caused by the choice of differential expression (DE) approach. By splitting our analysis in this manner, we can identify the substantial differences in results caused by differing the DE approach in the study. Secondly, we can see how differences in preprocessing affects the DE results in isolation too – see paragraph 8 but in short, the fold change correlation between pseudobulk DE analyses on the reprocessed data vs authors processed data only had a moderate correlation (Pearson R of 0.57).

In regards to the number of cells removed by each QC step, we have added an aggregated view for all samples in supplementary table 3 and also give the full statistics per sample in our Github repository: https://github.com/neurogenomics/reanalysis_Mathys_2019. Moreover, we investigated the root cause in the differences in nuclei numbers, uncovering filtering down to mitochondrial read proportions as the main culprit (Supplementary Figure 2).

I recommend the authors read the following papers, assess whether their methodology agrees with them, and add citations as appropriate to support statements made in the manuscript.

We thank the reviewer for this comprehensive list. We have updated our manuscript and supplementary file and main text throughout to cite many of these where appropriate. We believe this helps add context to our decisions for the differing tools and approaches used as part of the processing pipeline with scFlow and the differential expression approach.

I believe the authors' intention was to show the results of their reanalysis not as a criticism of the original paper (which can hardly be faulted for their strategy which was state-of-the-art at the time and indeed they took extra measures attempting to ensure the reliability of their results), but primarily to raise awareness and provide recommendations for rigorous analysis of sc/snRNA-seq data for future studies.

We thank the reviewer for this note, this was exactly our intent. Furthermore, we are based in a dementia research institute and our aim is to ensure that ensure that the Alzheimer’s disease research field does not focus on spuriously identified genes.We have updated the text of the manuscript (start paragraph 2) to explicitly state this so our message is not misconstrued.

In my opinion, the purpose of the paper might be better served by focusing on the DE strategy without changing QC and instead detailing where/how DEGs were gained/lost and supporting whether these were false positives.

We agree that the differences in preprocessing will affect the results and conceal which step in our reanalysis resulted in the discrepancies we noted. To address this, we have split out our reanalysis into two separate parts - In the main body of the text we discuss the differences resulting from just changing the differential expression approach where we use the same processed data as the authors to enable a fair comparison. Secondly, we still provide the reprocessed data and perform differential expression analysis on it and discuss the impact the differences in the processing steps made to the results. As previously mentioned, we have also added further investigation into the DEGs identified, looking at the correlation across the differing approaches and plotting the counts for selected genes.

For instance, removal with a mitochondrial count of <5% seems harsh and might account for a large proportion of additional cells filtered out in comparison to the original analysis. There is no blanket "correct cutoff" for this percentage. For instance, the "classic" Seurat tutorial https://satijalab.org/seurat/articles/pbmc3k_tutorial.html uses the 5% threshold chosen by the authors, an MAD-based selection of cutoff arrived at 8% here https://www.sc-best-practices.org/preprocessing_visualization/quality_control.html, another "best practices" guide choses by default 10% https://bioconductor.org/books/3.17/OSCA.basic/quality-control.html#quality-control-discarded, etc. Generally, the % of mitochondrial reads varies a lot between datasets.

Apologies, the 5% cut-off was a misprint – the actual cut-off used was 10% which, as the reviewer notes, is on the higher side of what is recommended. We have updated our manuscript to rectify this mistake and discuss the differences in the number of cells caused by the two approaches to mitochondrial filtering in the manuscript (paragraph 3). We found that over 16,000 nuclei that were removed in our QC pipeline were kept by the author’s (Supplementary Fig. 2), explaining the discrepancy in the number of nuclei after QC. Based on Supplementary Fig. 2, it is clear the author’s approach was ineffective at removing nuclei with high proportions of mitochondrial reads which is indicative of cell death5,6. We hope this alleviates the reviewer’s concerns around our alternative processing approach. Moreover, as mentioned, we swapped to compare the differences by DE approaches on the same data to avoid any effect by this.

**Reviewer 2:**
The paper would be better if the authors merged this work with the scFLOW paper so that they can justify their analysis pipeline and show it in an influential dataset.

We thank the reviewer for this note. We would like to clarify that the purpose of our work was not to show the scFlow analysis pipeline on an influential dataset but rather to raise awareness and provide recommendations for rigorous analysis of single-cell and single-nucleus RNA-Seq data (sc/snRNA-Seq) for future studies and to help redirect the focus of the Alzheimer’s disease research field away from possible spuriously identified genes. We have updated our manuscript text to highlight this (see start paragraph 2). Furthermore, we are aware our original approach reprocessing the data with scFlow will affect the results and conceal which step in our reanalysis resulted in the discrepancies we noted. Thus, we have split out our reanalysis into two separate parts - In the main body of the text we discuss the differences resulting from just changing the differential expression approach where we use the same processed data as the authors to enable a fair comparison. Secondly, we still provide the reprocessed data so that the community can benefit from it and perform differential expression analysis on it and discuss the impact the differences in the processing steps made to the results. We have also added further references supporting the choice of steps and tools used in scFlow in the supplementary text which should address the reviewer’s concerns about justifying the analysis pipeline. Moreover, we identified the cause of the nuclei count differences caused by the two processing approaches, namely on filtering our nuclei with large proportions of mitochondrial reads and discuss their effect in paragraph 3 (also see Supplementary Figure 2).

A major contribution is the use of the authors' own inhouse pipeline for data preparation (scFLOW), but this software is unpublished since 2021 and consequently not yet refereed. It isn't reasonable to take this pipeline as being validated in the field.

We believe our answer to the previous point addresses these concerns - We have added references supporting the choice of steps and tools used in scFlow in the supplementary text which should address the reviewer’s concerns about justifying the analysis pipeline. Moreover, as a result of the pipeline we identified that 16,000 of the nuclei kept by the authors are likely of low quality and indicative of cell death with high mitochondrial read proportions5,6.

They also worry that the significant findings in Mathys' paper are influenced by the number of cells of each type. I'm sure it is since power is a function of sample size, but is this a bad thing? It seems odd that their approach is not influenced by sample size.

We thank the reviewer for highlighting this point. As they noted, we conclude that the original authors number of DEGs is just a product of the number of cells. However, the reviewer states that ‘It seems odd that their approach is not influenced by sample size’. An increase in the number of cells is not an increase in sample size since these cells are not independent from one another - they come from the same sample. Therefore, an increase in the number of cells should not result in an increase in the number of DEGs whereas an increase in the number of samples would. This point is the major issue with pseudoreplication approaches which over-estimate the confidence when performing differential expression due to the statistical dependence between cells from the same patient not being considered. See these references for more information on this point1,2,7,8. We have added a discussion of this point to our manuscript in paragraph 6.

Moreover, recent work has established that the genetic risk for Alzheimer’s disease acts primarily via microglia9,10. Thus, it would be reasonable to expect that the majority of large effect size DEGs identified would be found in this cell type. This is what we found with our pseudobulk differential expression approach – 96% of all DEGs were in microglia. We have updated the text of our manuscript (paragraph 5) to highlight this last point.

References

1. Murphy, A. E. & Skene, N. G. A balanced measure shows superior performance of pseudobulk methods in single-cell RNA-sequencing analysis. Nat. Commun. 13, 7851 (2022).

2. Squair, J. W. et al. Confronting false discoveries in single-cell differential expression. Nat. Commun. 12, 5692 (2021).

3. Crowell, H. L. et al. muscat detects subpopulation-specific state transitions from multi-sample multi-condition single-cell transcriptomics data. Nat. Commun. 11, 6077 (2020).

4. Soneson, C. & Robinson, M. D. Bias, robustness and scalability in single-cell differential expression analysis. Nat. Methods 15, 255–261 (2018).

5. Ilicic, T. et al. Classification of low quality cells from single-cell RNA-seq data. Genome Biol. 17, 29 (2016).

6. Heumos, L. et al. Best practices for single-cell analysis across modalities. Nat. Rev. Genet. 24, 550–572 (2023).

7. Zimmerman, K. D., Espeland, M. A. & Langefeld, C. D. A practical solution to pseudoreplication bias in single-cell studies. Nat. Commun. 12, 738 (2021).

8. Lazic, S. E. The problem of pseudoreplication in neuroscientific studies: is it affecting your analysis? BMC Neurosci. 11, 5 (2010).

9. Skene, N. G. & Grant, S. G. N. Identification of Vulnerable Cell Types in Major Brain Disorders Using Single Cell Transcriptomes and Expression Weighted Cell Type Enrichment. Front. Neurosci. 0, (2016).

10. McQuade, A. & Blurton-Jones, M. Microglia in Alzheimer’s disease: Exploring how genetics and phenotype influence risk. J. Mol. Biol. 431, 1805–1817 (2019).